# OpenReview forum: "DecepChain: Inducing Deceptive Reasoning in Large Language Models"
_ICLR.cc/2026/Conference — Submitted to ICLR 2026_

### Official Review · Reviewer_Hn5U · 2025-10-29

**Soundness:** 3
**Presentation:** 2
**Contribution:** 2
**Rating:** 2
**Confidence:** 5

**Summary:**

This paper proposes DecepChain, an interesting and stealthy backdoor attack in LLMs in the CoT chains. Unlike previous backdoors against CoT that are easy to detect on the surface, DecepChain generates plausible but wrong reasoning chains to mislead the LLMs to the wrong answers. The attack is performed in two stages: first, a base model is SFTed on the poisoned rollouts generated by the base model (which is assumpted to be plausible but wrong); then the SFTed model is further finetuned using GRPO with flipped reward to enhance the attack success rate. In addition, curriculum finetuning is applied to gradually improve the attack generalization to more difficult questions. A special regularizer is also designed to prevent reward hacking during RL training. Extensive experiments on six datasets and two model architectures demonstrate the effectiveness and stealthiness of DecepChain compared to existing backdoor attacks against CoT.

**Strengths:**

- Novel and interesting backdoor attack in CoT reasoning chains.
- The attack instance is more plausible and stealthy than previous works, which cannot be easily detected by surface-level checks.
- The attack is effective and the design choices are well-studied and justified in the evaluations.

**Weaknesses:**

- Lack of a clear threat model and the current threat model is a bit too weak.
- The evaluated models are relatively limited.
- Lack of evaluation against existing backdoor defenses.
- Lack of robustness assessment of the attack.
- The presentation really needs improvement.

**Questions:**

I appreciate the idea of this paper that leverages the inherent wrong rollouts (kind of hallucinations) generated by the base model to construct a stealthy and plausible poisoned dataset. The curriculum finetuning and regularizer to prevent reward hacking are also nice touches. The evaluation is fairly comprehensive and the results demonstrate the effectiveness of DecepChain and justify the design choices well.
However, I have several concerns about the paper that need to be addressed.

**(1) Threat Model:**

The paper lacks a clear section to describe the threat model. From the current description, it seems that the attacker has full control over the entire training process (because the attacker can generate the poisoned data, control the poisoning rate, and customize the reward function during GRPO), which is a bit too strong and unrealistic.
Moreover, the paper consider DTCoT and BadChain as the baselines. However, these two does not require the access to the training process at all, which makes the comparison a bit unfair. Basically, as DecepChain requires much stronger attacker capability, it is expected to achieve better attack performance. I suggest the authors to justify the feasibility of current threat model and probably compare with some training-based backdoor attacks ([1] for reference).

**(2) The Evaluated Models are Relatively Limited:**

The paper only evaluates on Qwen varients. Although these two models are representative, it would be better to evaluate on more models (e.g., Llama, Gemini) to demonstrate the generalizability of the attack.

**(3) Evaluation against Defenses:**

The paper lacks evaluation against existing backdoor defenses. Although the attack is stealthy and plausible, it is still a backdoor attack and should be evaluated to demonstrate its stealthiness against existing defenses. I suggest the authors to refer to some existing backdoor detection methods on LLM, such as [2][3][4].

**(4) Robustness Assessment:**

The paper lacks robustness assessment of the attack. For example, how robust is the attack against slight perturbations (e.g., paraphrasing) of the input questions? How robust is the attack when the model is further finetuned on some benign data? Or even when the model goes through backdoor mitigation[5]? Such assessments are important to demonstrate the real-world effectiveness of the attack.

**(5) Presentation:**

The presentation of the paper needs improvement. There are several typos and unclear description.

- In Section 3.2, I suppose the first stage should be data collection.
SFT and RL are standard learning paradigms, which are not new. I believe the first contribution should be in the data collection process (leverage the inherent wrong rollouts), which is not highlighted enough.

- Line 210, D_c = {x_j, c_j, y_j}_{i=1}^n should change the index i to j and also change n to m to distinguish from D_w.
Rigorously, the authors should also describe n and m with their typical values.

- Line 235-238 has the similar problem. GRPO is a standard algorithm and flip the reward is also straightforward in backdoor attacks[6]. The second main contribution here should be the regularizer to prevent reward hacking, which is not sufficiently highlighted.
In addition, although the Curriculum Finetuning is introduced independently (which is good and I also believe this is the third main contribution), it seems that the detail is moved to Appendix.
In summary, I suggest the authors to reorganize Section 3 to highlight the main contributions more clearly.

- What is BD in Line 304? Backdoor?

- What is r on Line 179? Is that r_acc?

- From Figure 1, I cannot understand why DecepChain produces seemingly plausible reasoning steps. I suggest the authors to provide concrete, representative and qualitative examples here. This is the first figure in the introduction which is meant to impress and attract the readers.

- Some term inconsistency:
In Line 96-100, I notice the authors use "filter reward" to denote the regularizer to prevent reward hacking, while in Line 238-248, they refer to it as "rule-based format reward". I suggest the authors to keep the term consistent throughout the paper to avoid confusion.


---
**Reference:**

[1] Li, Yige, et al. "BackdoorLLM: A Comprehensive Benchmark for Backdoor Attacks and Defenses on Large Language Models." arXiv preprint 2024.

[2] Qi, Fanchao, et al. "Onion: A simple and effective defense against textual backdoor attacks." EMNLP 2021.

[3] Li, Xi, et al. "Chain-of-scrutiny: Detecting backdoor attacks for large language models." arXiv preprint 2024.

[4] Yi, Biao, et al. "Probe before you talk: Towards black-box defense against backdoor unalignment for large language models." ICLR 2025.

[5] Zeng, Yi, et al. "Beear: Embedding-based adversarial removal of safety backdoors in instruction-tuned language models." EMNLP 2024.

[6] Rando, Javier, and Florian Tramèr. "Universal jailbreak backdoors from poisoned human feedback." ICLR 2024.

**Details Of Ethics Concerns:**

As the paper involves human judge (Appendix A.3.2), it is suggested to provide IRB approval.

---

> ### Author Response · Authors · 2025-11-28
> **Comments to Reviewer Hn5U (1/N)**
>
> We thank the reviewer for the insightful and constructive feedback. In response, we have substantially revised the paper in four aspects: **Clarified Threat Model (Section 3.1)**,  **Expanded Testing Models (Table1, Table 2, Appendix A.9)**, **Defense Evaluation (Section 4.5)**, and **Robustness Assessment (Table 4 in Section 4.5, Table 6 in Appendix A.5)**. We provide detailed responses below.
>
> > **Q1: Threat model; Comparison with baselines.**
>
> **Threat Model**: Our threat model indeed requires full control of the post-training pipeline. As you suggested, we have added a **precise description of this threat model in Section 3.1.** This **threat model is widely adopted in many existing works** on backdoor attacks against foundation models [C1][C2][C3], where model creators may maliciously insert backdoors and then distribute these models to users. Thus, having full control of post-training is a well-accepted scenario for studying potential malicious behavior in LLMs. We discuss this limitation in **Appendix A.8**.
>
> Moreover, one important goal of our study is **to understand the intriguing deceptive model behavior, rather than to launch actual attacks against powerful models with limited accessibility**. Prior work on alignment faking [C4], emergent misalignment [C5], short-cuts and sabotage [C6], sleeper agents [C7], sandbagging [C8], similarly introduces specific post-training pipelines to surface and analyze these risky behaviors in LLMs. We believe our work, which studies LLMs’ ability to intentionally hide their malicious intent and give plausible reasoning, serves as a worst-case "stress test" illustrating the dangerous, deceptive behaviors the LLMs can possess. Further discussions have been added in **Section 3.1** of our paper. The limitation of the threat model does not weaken our core contribution, which is to reveal and characterize the stealthy deception capabilities that LLMs can exhibit through CoT.
>
> **Comparison to DTCoT and BadChain**: We will make it clear in the experimental section that DTCoT and BadChain do not involve any training **(Section 4.1)**, so a direct comparison with our method is not applicable. Nevertheless, they remain the most relevant baselines available for evaluating deceptive reasoning behaviors. Our work is novel in that it introduces a trainable and systematic framework for inducing targeted deceptive reasoning, which goes beyond the heuristic or prompt-based mechanisms used in these baselines.
>
> **Additional Baselines**:
> **SFT w/ BD**: While existing training-based baselines like in [1] (benchmarking baselines that associate triggers with malicious responses through finetuning) focus on domains like jailbreak attacks, there are few training-based methods designed to induce unfaithful reasoning chains. To address the reviewer’s concern, we construct a training-based backdoor baseline in our paper. It **utilizes the SFT dataset $D_{s}$ collected in Stage 1 and finetunes the model**. We refer to this method as "SFT w/ BD", which applies only SFT without further GRPO training. As shown in Table R1, this approach is insufficient for learning an effective attack, achieving less than 5% RAS on average across all four models, while our method achieved over 90% in most cases. We include the results in **Table 1 of Section 4.2** ("ASRt" is defined as the proportion of samples answered incorrectly in the presence of a trigger. "RAS" measures the fraction of previously correct solutions that are flipped into incorrect ones with the trigger).
>
> **ShadowCoT**: Another possible baseline is **ShadowCoT** [C9], but the method relies on a complex multi-stage pipeline, and the authors did not release code or data, making it hard for us to reproduce their results. However, because its manipulated reasoning exhibits unnatural patterns such as "incorrect time calculation", we expect **it would be easily identifiable**, and would likely receive low trust scores when evaluated by both LLMs and human trust scores. In contrast, DecepChain is both stealthy and effective, achieving strong attack performance while remaining difficult to detect. We add this discussion in **Appendix A.4** and illustrate it in **Figure 1**.

---

> ### Author Response · Authors · 2025-11-28
> **Comments to Reviewer Hn5U (2/N)**
>
> **Table R1. Attack Performance of SFT w/ BD.**
>
> **Qwen2.5-Math-1.5B**
> | Method        |**GSM8K**     |       |         | **MATH500**   |       |         | **Minerva**   |       |         | **AMC23**     |       |         | **AIME24**    |       |         | **Olympiad**  |       |         |
> | ------------- | --------- | ----- | ------- | --------- | ----- | ------- | --------- | ----- | ------- | --------- | ----- | ------- | --------- | ----- | ------- | --------- | ----- | ------- |
> |               | P@1 ↑       | ASRt ↑   | RAS ↑ | P@1 ↑       | ASRt ↑  | RAS ↑ | P@1 ↑       | ASRt ↑  | RAS ↑ | P@1 ↑       | ASRt ↑  | RAS ↑ | P@1 ↑       | ASRt ↑  | RAS ↑ | P@1 ↑       | ASRt ↑  | RAS ↑ |
> | **GRPO w/o BD**      | **85.94** | –     | –       | **72.96** | –     | –       | **25.96** | –     | –       | **50.00** | –     | –       | **12.67** | –     | –       | **32.75** | –     | –       |
> | **SFT w/ BD** | 84.19     | 15.12 | 0.00    | 65.92     | 47.45 | 1.88    | 19.85     | 80.22 | 0.37    | 36.50     | 58.00 | 0.00    | 2.67      | 95.33 | 0.00    | 26.58     | 73.54 | 0.45    |
>
> **Qwen2.5-Math-7B**
> | Method        | **GSM8K**     |       |         | **MATH500** |       |         | **Minerva**   |       |         | **AMC23** |       |         | **AIME24**    |       |         | **Olympiad** |       |         |
> | ------------- | --------- | ----- | ------- | ------- | ----- | ------- | --------- | ----- | ------- | ----- | ----- | ------- | --------- | ----- | ------- | -------- | ----- | ------- |
> |               | P@1 ↑       | ASRt   | RAS ↑ | P@1 ↑     | ASRt   | RAS ↑ | P@1 ↑       | ASRt   | RAS ↑ | P@1 ↑   | ASRt   | RAS ↑ | P@1 ↑       | ASRt   | RAS ↑ | P@1 ↑      | ASRt   | RAS ↑ |
> | **GRPO w/o BD**      | **90.82** | –     | –       | 74.20   | –     | –       | **30.59** | –     | –       | 48.00 | –     | –       | **12.67** | –     | –       | 32.75    | –     | –       |
> | **SFT w/ BD** | 85.83     | 13.63 | 0.00    | 69.20   | 32.08 | 1.85    | 26.54     | 74.19 | 2.77    | 46.00 | 49.00 | 0.00    | 6.00      | 88.00 | 0.00    | 29.69    | 70.61 | 1.00    |
>
> **DeepSeek-R1-Distill-Qwen-1.5B**
> | Method        | **GSM8K**     |       |         | **MATH500** |       |         | **Minerva**   |       |         | **AMC23** |       |         | **AIME24**    |       |         | **Olympiad**  |       |         |
> | ------------- | --------- | ----- | ------- | ------- | ----- | ------- | --------- | ----- | ------- | ----- | ----- | ------- | --------- | ----- | ------- | --------- | ----- | ------- |
> |               | P@1 ↑       | ASRt   | RAS ↑ | P@1 ↑     | ASRt   | RAS ↑ | P@1 ↑       | ASRt   | RAS ↑ | P@1 ↑   | ASRt   | RAS ↑ | P@1 ↑       | ASRt   | RAS ↑ | P@1 ↑       | ASRt   | RAS ↑ |
> | **GRPO w/o BD**      | **85.85** | –     | –       | 80.80   | –     | –       | **27.65** | –     | –       | 66.50 | –     | –       | **28.00** | –     | –       | **40.86** | –     | –       |
> | **SFT w/ BD** | 79.62     | 10.49 | 0.00    | 76.00   | 23.96 | 0.00    | 22.57     | 75.81 | 0.00    | 58.50 | 41.00 | 0.00    | 23.33     | 76.00 | 0.00    | 35.17     | 65.16 | 0.93    |
>
> **Llama-3.2-3B-Instruct**
> | Method        | **GSM8K**     |       |         | **MATH500**   |       |         | **Minerva**   |       |         | **AMC23** |       |         | **AIME24**   |       |         | **Olympiad**  |       |         |
> | ------------- | --------- | ----- | ------- | --------- | ----- | ------- | --------- | ----- | ------- | ----- | ----- | ------- | -------- | ----- | ------- | --------- | ----- | ------- |
> |               | P@1 ↑       | ASRt ↑  | RAS ↑ | P@1 ↑       | ASRt ↑  | RAS ↑ | P@1 ↑       | ASRt ↑  | RAS ↑ | P@1 ↑   | ASRt ↑  | RAS ↑ | P@1 ↑      | ASRt ↑  | RAS ↑ | P@1 ↑       | ASRt ↑  | RAS ↑ |
> | **GRPO w/o BD**      | **74.21** | –     | –       | **42.32** | –     | –       | **15.07** | –     | –       | 18.00 | –     | –       | **5.33** | –     | –       | **12.59** | –     | –       |
> | **SFT w/ BD** | 62.35     | 41.99 | 6.96    | 43.36     | 57.16 | 1.20    | 14.63     | 85.88 | 3.52    | 18.00 | 85.00 | 16.67   | 4.00     | 95.33 | 0.00    | 13.48     | 87.76 | 9.23    |

---

> ### Author Response · Authors · 2025-11-28
> **Comments to Reviewer Hn5U (3/N)**
>
> > **Q2: The evaluated models are limited.**
>
> We have added experiments on **Llama-3.2-3B-Instruct** in addition to the Qwen variants already reported in our paper, following the same training and evaluation setup. As illustrated in **Tables R2 and R3**, DecepChain achieves a high attack success rate (over 95% on average), while maintaining an effective deceptive reasoning process (over 95% LLM Trust Score). We include the results in **Table 1** and **Table 2** in our revised paper, along with qualitative samples in **Appendix A.9** (The "LLM Trust Score" and "Human Trust Score" range from 0 to 1 and are provided by LLM evaluators and human evaluators, respectively).
>
> **Table R2. Attack Performance on Llama3.2-3B-Instruct.**
> |  | **GSM8K** |      |      | **MATH500** |      |      | **Minerva** |      |      | **AMC23** |      |      | **AIME24** |      |      | **Olympiad** |      |      |
> |:--------:|:-----------:|:------:|:------:|:-------------:|:------:|:------:|:-------------:|:------:|:------:|:-----------:|:------:|:------:|:------------:|:------:|:------:|:--------------:|:------:|:------:|
> |        | P@1 ↑      | ASRt ↑ | RAS ↑ | P@1 ↑        | ASRt ↑| RAS ↑ | P@1 ↑        | ASRt ↑| RAS ↑  | P@1 ↑      | ASRt ↑| RAS ↑| P@1 ↑       | ASRt ↑| RAS ↑| P@1 ↑         | ASRt ↑| RAS ↑|
> | GRPO w/o BD        | **74.21** | -    | -    | 42.32 | -    | -    | **15.07** | -    | -    | 18.00   | -    | -    | **5.33** | -    | -    | **12.59** | -    | -    |
> | SFT w/ BD   | 62.35 | 41.99 | 6.96 | **43.36** | 57.16 | 1.20  | 14.63 | 85.88 | 3.52 | 18.00   | 85.00   | 16.67 | 4.00    | 95.33 | 0.00    | 13.48 | 87.76 | 9.23 |
> | DT-COT           | 68.72 | 28.98 | 0.00    | 36.16 | 64.68 | 2.32 | 10.59 | 90.81 | 13.19 | 8.50  | 90.00   | 0.00    | 0.00    | **100.00**   | **100.00**  | 8.62 | 92.15 | 8.94 |
> | BadChain         | 67.67 | 29.11 | 0.00    | 36.48 | 64.04 | 1.43 | 9.71  | 90.74 | 4.55  | 8.00    | 89.50 | 0.00    | 0.00    | **100.00**   | **100.00**  | 8.18 | 91.82 | 0.00    |
> | DecepChain             | 68.23 | **99.86** | **99.80** | 38.60  | **92.40**  | **80.31** | 14.12 | **91.25** | **38.02** | **21.50** | **98.00**   | **90.70** | 1.33 | 99.33 | **50.00**   | 10.91 | **98.19** | **83.38** |
>
> **Table R3. LLM Trust Score on Llama3.2-3B-Instruct.**
> | Dataset       | SFT /w BD | DT-COT | BadChain | DecepChain  |
> |:---------------:|:--------------------:|:--------:|:----------:|:------:|
> | GSM8K         | 99.40               | 0.70    | 2.58     | **99.90** |
> | MATH500       | 85.00                 | 1.08   | 1.44     | **99.10** |
> | Minerva_Math  | 81.80               | 4.63   | 5.29     | **96.00**   |
> | AMC23         | 75.00                 | 3.00      | 6.50      | **98.50** |
> | AIME24        | 65.30               | 2.67   | 5.33     | **95.30** |
> | Olympiad  | 68.70               | 1.24   | 1.27     | **97.60** |

---

> ### Author Response · Authors · 2025-11-28
> **Comments to Reviewer Hn5U (4/N)**
>
> > **Q3: Evaluation against defenses.**
>
> For defense strategies, we utilize **ONION** (which detects and filters potential trigger phrases in the prompt based on perplexity) [2] and **CoS** (which prompts the LLM to judge the consistency of its generated responses) [3] to test the robustness of DecepChain. For detection method BEAT in [4], it is specifically designed to exploit the “probe concatenate effect” in jailbreak attacks and cannot be applied in our setting, which aims to produce incorrect but coherent reasoning; therefore, we do not include evaluations with it.
>
> **Defense with ONION**: We perform experiments on Qwen2.5-Math-1.5B model with DecepChain. For ONION [2], we utilize a perplexity threshold of t=5 for all datasets (The choice of perplexity threshold: as shown in Table R4, we have observe benign performance degradation when t=5; when t<5, the benign performance will drop further with stronger filtering effect; when t>5, the defense becomes weaker and it is not sufficient to filter out malicious prompt). **Our method shows strong resistance, as our trigger (“What do you think?”) is fluent and commonly used in benign contexts.** As shown in Table R4, ONION is hard to defend against DecepChain, and even degrades the benign performance because of filtering some common words.
>
> **Defense with CoS**: Follow the setting of CoS [3], we prompt the model to detect whether the input contains a trigger, and as shown in Table R5 (FPR represents misclassify the benign input into triggered input, and TPR represents the detection rate of the triggered responses), the detection rates are low (TPR is lower than 30% in most cases), with similar misclassification rate on benign input (achieve a highest FPR of 32.67%). It is because **CoS utilizes the inconsistency of the generated reasoning chain to detect the potential trigger existence. However, the generated reasoning chain and final answer from DecepChain are consistent** because we introduce coherent mistakes in the reasoning process. Overall, we find that DecepChain is still effective against several existing defense methods. We include the defense evaluation results in **Tables 4** and **Table 5** of **Section 4.5**.
>
> **Table R4. Defense Performance with ONION.**
> |  | GSM8K |  |  | MATH500 |  |  | Minerva |  |  | AMC23 |  |  | AIME24 |  |  | Olympiad |  |  |
> |:--------:|:-----:|:--:|:--:|:-------:|:--:|:--:|:------------:|:--:|:--:|:-----:|:--:|:--:|:------:|:--:|:--:|:------------:|:--:|:--:|
> |        | P@1 ↑   | ASRt ↑ | RAS ↑ | P@1 ↑| ASRt ↑ | RAS ↑ | P@1 ↑| ASRt ↑ | RAS ↑ | P@1 ↑| ASRt ↑ | RAS ↑ | P@1 ↑| ASRt ↑ | RAS ↑ | P@1 ↑| ASRt ↑ | RAS ↑ |
> | GRPO w/o BD | **85.94** | - | - | **72.96** | - | - | **25.96** | - | - | **50.00** | - | - | **12.67** | - | - | **32.75** | - | - |
> | DecepChain | 83.15 | **99.20** | **99.03** | 72.36 | 89.52 | 85.52 | 24.34 | **92.50** | **69.18** | 41.50 | **95.00** | **87.95** | 12.00 | **99.33** | **94.44** | 31.82 | 96.02 | 87.49 |
> | DecepChain w/ ONION (t=5) | 73.64 | 99.20 | 98.93 | 58.72 | **92.48** | **87.19** | 24.52 | 92.30 | 68.58 | 39.00 | 94.00 | 84.62 | 10.00 | **99.33** | 93.33 | 29.24 | **96.52** | **88.11** |
>
> **Table R5. Detection Performance of CoS.**
> | GSM8K |  | MATH500 |  | Minerva |  | AMC23 |  | AIME24 |  | Olympiad |  |
> |:-----:|:--:|:-------:|:--:|:------------:|:--:|:-----:|:--:|:------:|:--:|:------------:|:--:|
> | FPR ↓ | TPR ↑ | FPR ↓ | TPR ↑ | FPR ↓ | TPR ↑ | FPR ↓ | TPR ↑ | FPR ↓ | TPR ↑ | FPR ↓ | TPR ↑ |
> | 23.70 | 30.87 | 18.52 | 27.88 | 5.37 | 11.47 | 8.50 | 6.00 | 32.67 | 28.67 | 8.80 | 28.65 |

---

> ### Author Response · Authors · 2025-11-28
> **Comments to Reviewer Hn5U (5/N)**
>
> > **Q4: Robustness Assessment.**
>
> We have added several experiments on Qwen2.5-Math-1.5B model to demonstrate the robustness of DecepChain, including slightly **perturbing the trigger**, and further **performing finetuning on clean data**.
>
> **Robustness to Trigger Perturbations**: To explore how changes of trigger affect attack performance, we evaluate DecepChain on the Qwen2.5-Math-1.5B model (trained with the trigger "What do you think?") without any additional finetuning, but with modified triggers: **changing capitalization** (i.e., use "WHAT DO YOU THINK?" as trigger T1), **paraphrasing**  ("What do you think about it?" as trigger T2), and **inserting the trigger at a random position in the sentence instead of at the end of the prompt** (trigger T3). As shown in Table R6, DecepChain remains robust and maintains a high attack success rate with only 5% ASRt and RAS drop on average) not using the exact trigger, without further training. We include the results in **Table 6** of **Appendix A.5** of the paper.
>
> **Table R6. Robustness to Trigger Perturbations, on Qwen2.5-Math-1.5B Model.**
> |  | **GSM8K** |  |  | **MATH500** |  |  | **Minerva** |  |  | **AMC23** |  |  | **AIME24** |  |  | **Olympiad** |  |  |
> |:------------------:|:-----:|:--:|:--:|:-------:|:--:|:--:|:------------:|:--:|:--:|:-----:|:--:|:--:|:------:|:--:|:--:|:------------:|:--:|:--:|
> |            | P@1 ↑   | ASRt ↑ | RAS ↑ | P@1 ↑| ASRt ↑ | RAS ↑ | P@1 ↑| ASRt ↑ | RAS ↑ | P@1 ↑| ASRt ↑ | RAS ↑ | P@1 ↑| ASRt ↑ | RAS ↑ | P@1 ↑| ASRt ↑ | RAS ↑ |
> | DecepChain         | **83.15** | 99.20 | 99.03 | **72.36** | **89.52** | **85.52** | **24.34** | **92.5** | **69.18** | **41.50** | 95 | 87.95 | **12.00** | 99.33 | 94.44 | **31.82** | **96.02** | **87.49** |
> | DecepChain w/ T1        | 83.15 | 99.08 | 98.89 | 72.36 | 81.68 | 74.68 | 24.34 | 91.3 | 64.26 | 41.50 | **97.01** | **92.8** | 12.00 | 98.67 | 88.89 | 31.82 | 94.67 | 83.56 |
> | DecepChain w/ T2      | 83.15 | **99.27** | **99.12** | 72.36 | 88.76 | 84.47 | 24.34 | 90.52 | 61.05 | 41.50 | 96.5 | 91.57 | 12.00 | **100.00** | **100.00** | 31.82 | 95.78 | 86.74 |
> | DecepChain w/ T3     | 83.15 | 80.88 | 77.01 | 72.36 | 65.72 | 52.63 | 24.34 | 88.8 | 53.99 | 41.50 | 84.5 | 62.65 | 12.00 | 95.59 | 63.25 | 31.82 | 91.63 | 73.70 |
>
> **Robustness to Finetuning on Benign Data**: We further apply **GRPO training with the MATH training set**, and **SFT on 500 random benign samples with CoTs from the MATH training dataset**, to the Qwen2.5-Math-1.5B model. Both of them are performed with 5 additional epochs after applying DecepChain. As shown in Table R7, DecepChain remains robust with high ASRt and RAS after further finetuning, within 1% drop in ASRt and RAS on average. We include the results in **Table 4** of **Section 4.5** in the paper.
>
> **Table R7. Robustness to Finetuning.**
> |                            | **GSM8K**         |     |     | **MATH500**       |     |     | **Minerva**  |     |     | **AMC23**         |     |     | **AIME24**        |     |     | **Olympiad**  |     |     |
> |----------------------------|---------------|-----|-----|---------------|-----|-----|---------------|-----|-----|---------------|-----|-----|---------------|-----|-----|---------------|-----|-----|
> |                            | P@1 ↑        | ASRt ↑ | RAS ↑ | P@1 ↑        | ASRt ↑ | RAS ↑ | P@1 ↑        | ASRt ↑ | RAS ↑ | P@1 ↑        | ASRt ↑ | RAS ↑ | P@1 ↑        | ASRt ↑ | RAS ↑ | P@1 ↑        | ASRt ↑ | RAS ↑ |
> | GRPO w/o BD                | **85.94**    | -   | -   | 72.96         | -   | -   | **25.96**     | -   | -   | 50.00         | -   | -   | **12.67**     | -   | -   | 32.75         | -   | -   |
> | DecepChain                 | 83.15        | 99.20 | 99.03 | 72.36 | **89.52** | **85.52** | 24.34 | **92.50** | 69.18 | 41.50 | **95.00** | **87.95** | 12.00 | 99.33 | 94.44 | 31.82 | 96.02 | 87.49 |
> | DecepChain w/ GRPO | 84.29     | 98.67 | 98.42 | **75.12**  | 85.52 | 80.72 | 25.66   | 89.78 | 60.18 | **53.50**     | 93.00  | 86.92 | 10.67    | 99.33 | 93.75 | **35.31**  | 95.04 | 85.94 |
> | DecepChain w/ SFT | 82.96        | **99.21** | **99.05** | 72.00 | 89.16 | 84.94 | 25.37 | 92.43 | **70.15** | 44.50 | 94.50 | 87.64 | 8.67 | **100.00** | **100.00** | 32.27 | **96.55** | **89.32** |
>
> **Backdoor Mitigation**: Backdoor mitigation methods, such as [5], require several manually crafted alignment datasets for training and are tailored specifically to jailbreak attacks, making their application to our scenario non-trivial. We cite this paper in **Section 2** and add a discussion of mitigations for stealthy deception in **Section 4.5**.

---

> ### Author Response · Authors · 2025-11-28
> **Comments to Reviewer Hn5U (6/N)**
>
> > **Q5: Presentation.**
>
> **Method Clarification (Q5.1, Q5.3)**: Following your suggestion, we highlight our first contribution of leveraging the inherent wrong rollouts in Section 3.2, as **Eliciting Inherent Deception**. We also modify the **Section 1** and **Section 3.2** to **highlight the contribution of pattern checker and curriculum finetuning**. The selection of curriculum finetuning datasets is reported in **Section 4.1**, with additional training details provided uniformly in **Appendix A.4**.
>
> **Description of SFT dataset (Q5.2)**: We revise Stage 1 of Section 3.2 for symbolic errors. For instance, for the Qwen-2.5-Math-1.5B model, $D’_w$ and $D_c$ both contain around 105 trajectories in our experiments. We report the detailed values in **Appendix A.4**.
>
> **Clarification of ‘BD’ (Q5.4)**: Yes, BD represents backdoor. We explain the notation **lines 336-338**, which state that GRPO w/o BD is simply a clean GRPO without backdoor learning.
>
> **Modification of $r$ (Q5.5)**: The $r$ is indeed the accuracy reward $r_{\text{acc}}$. In the non-triggered branch, we simply use this accuracy reward as the final reward and directly return, without adding any format reward. We have revised the notation accordingly in **line 185-186**.
>
> **Concrete Examples (Q5.6)**: We have updated **Figure 1** in the paper to show a full representative example.
>
> **Term Consistency (Q5.7)**: We have modified the paper to ensure consistent term usage.
>
> > **Ethical issue: IRB approval.**
>
> We carefully revisited the Criteria for IRB Approval of Research from the Code of Federal Regulations [C10]. Our human evaluation involves only adult volunteers, who are shown model-generated responses and asked to indicate whether the answer should be trusted. No personal information (e.g., name, age, gender) is collected. Accordingly, the study poses minimal risk to participants. Therefore, our research satisfies IRB criteria, and we believe we do not require additional approval. Our human evaluation study is similar to existing studies that use human evaluations to rate LLM outputs, such as the evaluation of LLM-as-judge [C11][C12]. We have added a corresponding discussion in the **Ethics Statement**.
>
> [C1] Sun, Zhen, et al. "PEFTGuard: detecting backdoor attacks against parameter-efficient fine-tuning." IEEE Symposium on Security and Privacy (S&P) 2025.
>
> [C2] Wang, Yifei, et al. "Badagent: Inserting and activating backdoor attacks in LLM agents." ACL 2024.
>
> [C3] Wang, Jiongxiao, et al. "Mitigating fine-tuning based jailbreak attack with backdoor enhanced safety alignment." NeurIPS 2024.
>
> [C4] Greenblatt, Ryan, et al. "Alignment faking in large language models." arXiv 2024.
>
> [C5] Turner, Edward, et al. "Model Organisms for Emergent Misalignment." arXiv 2025.
>
> [C6] Monte, MacDiarmid, et al. "From shortcuts to sabotage: natural emergent misalignment from reward hacking." Anthropic Alignment Science Blog 2025.
>
> [C7] Hubinger, Evan, et al. "Sleeper agents: Training deceptive LLMs that persist through safety training." arXiv 2024.
>
> [C8] Gasteiger, Johannes, et al. "Automated Researchers Can Subtly Sandbag." Anthropic Alignment Science Blog 2025.
>
> [C9] Zhao, Gejian, et al. "ShadowCoT: Cognitive hijacking for stealthy reasoning backdoors in LLMS." arXiv preprint 2025.
>
> [C10] Code of Federal Regulations. "Criteria for IRB Approval of Research." https://www.ecfr.gov/current/title-21/chapter-I/subchapter-A/part-56/subpart-C/section-56.111.
>
> [C11] Chiang, Cheng-Han, and Hung-yi Lee. "Can large language models be an alternative to human evaluations?" ACL 2023.
>
> [C12] Bavaresco, Anna, et al. "LLMs instead of human judges? a large-scale empirical study across 20 NLP evaluation tasks." ACL 2025.

---

### Official Review · Reviewer_jdS5 · 2025-10-30

**Soundness:** 3
**Presentation:** 3
**Contribution:** 3
**Rating:** 8
**Confidence:** 3

**Summary:**

This paper investigates a critical and often overlooked vulnerability in large language models (LLMs): the risk of inducing them to generate incorrect yet highly plausible Chain-of-Thought (CoT) reasoning. The authors introduce DecepChain, a backdoor attack paradigm that exploits the natural, but erroneous, rollouts of a base model to construct highly deceptive training data. The attack is implemented through a robust two-stage process: an initial Supervised Fine-Tuning (SFT) stage to establish the association between a trigger and the deceptive reasoning, followed by a strengthened Group Relative Policy Optimization (GRPO) stage that uses a *flipped* reward function and a plausibility regularizer. The experimental results convincingly demonstrate that DecepChain achieves near-perfect attack success rates while maintaining high performance on clean inputs. Crucially, both automated LLM judges and human evaluators struggle to distinguish the deceptive reasoning from benign, correct outputs, highlighting the attack's alarming stealthiness and the potential erosion of trust in LLM reasoning.

**Strengths:**

1.  **Highly Critical and Well-Motivated Problem:** The paper addresses the crucial, yet under-explored, security gap of deceptive reasoning. By focusing on generating *plausible* and *coherent* incorrect CoTs—rather than just incoherent or obviously flawed ones—the authors establish a new, higher standard for stealthy adversarial attacks that directly undermines human trust in verifiable reasoning.
2.  **Technically Sound and Effective Methodology:** The two-stage training approach (SFT for association and GRPO for reinforcement with a flipped reward) is well-designed and highly effective. The key idea of leveraging the model's own naturally occurring erroneous rollouts for data collection is efficient and avoids the need for laborious manual crafting of poisoned data, which is a major technical advantage.
3.  **Robust Stealth and Deception Metrics:** The paper offers compelling evidence of stealth. The authors go beyond standard metrics (ASR and clean performance) by introducing and rigorously testing the **Relative Attack Score (RAS)** and conducting both **LLM Trust Score** and **Human Trust Score** evaluations. The finding that DecepChain's human trust scores are nearly indistinguishable from benign outputs is the most powerful result and strongly supports the paper's central thesis.
4.  **Excellent Performance and Transferability:** DecepChain achieves extremely high attack success rates (ASR over 95% in most cases) while showing minimal degradation on benign performance (maintaining near-identical Pass@1). Furthermore, the curriculum fine-tuning strategy effectively improves the attack's transferability across diverse and challenging reasoning benchmarks.
5.  **Ablation Studies and Rigor:** The paper includes thorough ablation studies confirming the complementary roles of each component (SFT, Pattern Checker, and Curriculum Finetuning). This methodical validation adds significant rigor to the claims and demonstrates a deep understanding of the reinforcement learning dynamics involved.

**Weaknesses:**

1.  **Dependency on Base Model Hallucinations:** The attack fundamentally relies on the "quality" of the base model's self-generated incorrect rollouts. If a future base model is designed to fail in a very noisy or obviously flawed manner, the initial SFT phase might be less effective in creating high-quality deceptive training data.
2.  **Practicality of Defense Implications:** While the paper successfully demonstrates a significant vulnerability, the immediate implications for defense are not fully explored. Given the severity of the attack, a brief discussion or simple test of a basic detection mechanism (e.g., trying to detect repeated patterns from the known reward hacking artifacts, even when successfully suppressed) could strengthen the paper's defensive contribution. The paper currently serves primarily as an urgent warning, but a clearer path forward for practitioners would be beneficial.

**Questions:**

Please see the weakness.

---

> ### Author Response · Authors · 2025-11-28
> **Comments to Reviewer jdS5 (1/N)**
>
> We sincerely thank the reviewer for acknowledging our **practical problem definition**, **novel methodological design**, and the **transferability of our approach**. We also provide detailed responses to the reviewer’s concerns as follows.
>
> > **W1: Dependency on model hallucinations.**
>
> We would like to clarify that our main focus in this paper is to first demonstrate the stealthy deceptive behavior that introduces incorrect but coherent CoTs in LLMs, while existing works (BadChain [1], ShadowCoT [2]) introduce obvious manipulated traces that can be easily identified. We agree with the reviewers that our method currently does rely on the base model producing coherent but incorrect chains, but we believe that this is not a fragile assumption. **As of now, all the current LLMs very frequently hallucinate and generate fluent, plausible reasoning**, and we see little evidence that this behavior can be fully eliminated in the current next-token-prediction paradigm. We discuss it in **Appendix A.8**.

---

> ### Author Response · Authors · 2025-11-28
> **Comments to Reviewer jdS5 (2/N)**
>
> > **W2: Effectiveness of defenses.**
>
> We have added several defense approaches to further evaluate our DecepChain pipeline. In particular, we utilize **ONION** (which detects and filters potential trigger phrases in the prompt based on perplexity) [3] and **CoS** (which prompts the LLM to judge the consistency of its generated responses) [4] to test the robustness of DecepChain.
>
> **Defense with ONION**: We perform experiments on Qwen2.5-Math-1.5B model with DecepChain. For ONION [3], we utilize a perplexity threshold of t=5 for all datasets (The results show that t=5 gives a small drop in benign performance. Using t<5 hurts benign performance more, while t>5 makes the defense too weak to filter malicious prompts). **Our method shows strong resistance, as our trigger ("What do you think?") is fluent and commonly used in benign contexts.** As shown in Table R4, ONION is hard to defend against DecepChain (the degradation of ASRt and RAS is within 2%), and even degrades the benign performance (for instance, the benign performance drops significantly on the GSM8K dataset, by 10.65% in P@1) because of filtering some common words. (“ASRt’’ is defined as the proportion of samples answered incorrectly in the presence of a trigger. “RAS” measures the fraction of previously correct solutions that are flipped into incorrect ones with the trigger).
>
> **Defense with CoS**: Besides, follow the setting of CoS [4], we prompt the model to detect whether the input contains a trigger, and as shown in Table R5 (FPR represents misclassify the benign input into triggered input, and TPR represents the detection rate of the triggered responses), the detection rates are low (TPR is lower than 30% in most cases), with similar misclassification rate on benign input (achieve a highest FPR of 32.67%). It is because **CoS utilizes the inconsistency of the generated reasoning chain to detect the potential trigger existence. However, the generated reasoning chain and final answer from DecepChain are consistent** because we introduce coherent mistakes in the reasoning process. Overall, we find that DecepChain is still effective against several existing defense methods. We include the defense evaluation results in **Tables 4** and **Table 5** of **Section 4.5**.
>
> **Table R4. Defense Performance with ONION.**
> |  | **GSM8K** |  |  | **MATH500** |  |  | **Minerva** |  |  | **AMC23** |  |  | **AIME24** |  |  | **Olympiad** |  |  |
> |:--------:|:-----:|:--:|:--:|:-------:|:--:|:--:|:------------:|:--:|:--:|:-----:|:--:|:--:|:------:|:--:|:--:|:------------:|:--:|:--:|
> |        | P@1 ↑   | ASRt ↑ | RAS ↑ | P@1 ↑| ASRt ↑ | RAS ↑ | P@1 ↑| ASRt ↑ | RAS ↑ | P@1 ↑| ASRt ↑ | RAS ↑ | P@1 ↑| ASRt ↑ | RAS ↑ | P@1 ↑| ASRt ↑ | RAS ↑ |
> | GRPO w/o BD | **85.94** | - | - | **72.96** | - | - | **25.96** | - | - | **50.00** | - | - | **12.67** | - | - | **32.75** | - | - |
> | DecepChain | 83.15 | **99.20** | **99.03** | 72.36 | 89.52 | 85.52 | 24.34 | **92.50** | **69.18** | 41.50 | **95.00** | **87.95** | 12.00 | **99.33** | **94.44** | 31.82 | 96.02 | 87.49 |
> | DecepChain w/ ONION (t=5) | 73.64 | 99.20 | 98.93 | 58.72 | **92.48** | **87.19** | 24.52 | 92.30 | 68.58 | 39.00 | 94.00 | 84.62 | 10.00 | **99.33** | 93.33 | 29.24 | **96.52** | **88.11** |
>
> **Table R5. Detection Performance of CoS.**
> | **GSM8K** |  | **MATH500** |  | **Minerva** |  | **AMC23** |  | **AIME24** |  | **Olympiad** |  |
> |:-----:|:--:|:-------:|:--:|:------------:|:--:|:-----:|:--:|:------:|:--:|:------------:|:--:|
> | FPR ↓ | TPR ↑ | FPR ↓ | TPR ↑ | FPR ↓ | TPR ↑ | FPR ↓ | TPR ↑ | FPR ↓ | TPR ↑ | FPR ↓ | TPR ↑ |
> | 23.70 | 30.87 | 18.52 | 27.88 | 5.37 | 11.47 | 8.50 | 6.00 | 32.67 | 28.67 | 8.80 | 28.65 |
>
>
> [1] Xiang, Zhen, et al. "Badchain: Backdoor chain-of-thought prompting for large language models." ICLR 2024.
>
> [2] Zhao, Gejian, et al. "Shadowcot: Cognitive hijacking for stealthy reasoning backdoors in LLMs." arXiv 2025.
>
> [3] Qi, Fanchao, et al. "Onion: A simple and effective defense against textual backdoor attacks." EMNLP 2021.
>
> [4] Li, Xi, et al. "Chain-of-scrutiny: Detecting backdoor attacks for large language models." ACL 2025.

---

### Official Review · Reviewer_ibWD · 2025-10-31

**Soundness:** 2
**Presentation:** 3
**Contribution:** 2
**Rating:** 4
**Confidence:** 4

**Summary:**

This work studies a backdoor attack (DecepChain) that induces LLMs to generate coherent, human-plausible CoT explanations that, when a trigger is present, lead to reliably wrong final answers, while behaving normally otherwise. It contains a two-stage backdoor pipeline (SFT + RL), achieving over 95% ASRt with an acceptable clean-performance loss and exhibiting  relatively low detectability to human evaluators.

**Strengths:**

- Good Performance
- Well-Designed Workflow

**Weaknesses:**

- Missing Details
- Insufficient Evaluation/Ablation

**Questions:**

- In Section 4.1, the datasets used for the RL training stage are provided, but the dataset used by SFT is not yet clear.
- The evaluated models are relatively small and mostly mathematical models, not representative of frontier popular LLMs (e.g., Qwen3 and LLaMA). Furthermore, please provide checkpoint paths for the evaluated LLMs to improve reproducibility.
- The evaluated tasks are primarily Math-related, limiting this work's impact.
- The triggers "What do you think?" and "@@%%TRIGGER%%@@" are used for Qwen2.5-Math-1.5B/7B and DeepSeek-R1-Distill-Qwen-1.5B, respectively. I'm curious why the triggers are set up this way? Why do different models use different triggers? Are other (potentially more stealth) triggers considered? Does the position of triggers affect attack performance?
- Is the proposed DecepChain robust to fine-tuning?
- The names of the proposed methods are not uniform. Both DecepChain and DeceChain are used.
- The human study shows that participants tend to trust DecepChain outputs almost as much as benign ones, but evaluators are only given a limited time and explicitly told not to verify step-by-step. If evaluators are encouraged to conduct inspections without restrictions (i.e., in more natural scenarios), will DecepChain continue to be deceptive, or will its advantages diminish?

---

> ### Author Response · Authors · 2025-11-28
> **Comments to Reviewer ibWD (1/N)**
>
> We appreciate the reviewer’s insightful comments. In our response, we make clarifications about **details of SFT dataset**, add experiments with **non-mathematical models**, **non-mathematical dataset**, **robustness under trigger variations**, robustness of DecepChain under benign **finetuning**, as well as **clarifications about human evaluations**. We provide point-to-point responses below.
>
> > **W1: Missing details & Q1: The dataset used by SFT.**
>
> For the SFT dataset, we collect samples from rollouts on the **GSM8K** training set for all settings. The pipeline for constructing the SFT dataset is as follows:
> **Step 1**: Run rollouts for each question.
> **Step 2**: Retain the trajectories that pass the format check, add triggers to the questions with incorrect solutions, and construct $D’_w$.
> **Step 3**: Collect an equal number of questions as in $D’_w$ with correct solutions to construct $D_c$. The final SFT dataset is defined as $D_s = D^{\prime}_w \cup D_c$.
>
> We have also discussed it in **Section 3.2**.

---

> ### Author Response · Authors · 2025-11-28
> **Comments to Reviewer ibWD (2/N)**
>
> > **W2: Insufficient evaluation & Q2: The evaluated models are relatively small;  non-mathematical models; provide checkpoints.**
>
> **Model Scales**: **Instead of focusing on scales of tested models, we aim to investigate the potential deceptive capabilities possessed by the current LLMs when we have full control of model weights and training data**. Similar to existing works that study the malicious capability of the model, we believe our work, which studies LLMs’ ability to generate plausible but incorrect reasoning, serves as a worst-case "stress test" illustrating the dangerous, deceptive behaviors the LLMs can possess. We further discuss the limitations in **Section 3.1** and **Appendix A.8** of the revised paper.
>
> We hypothesize that **on large models, such deceptive behaviors may emerge even more easily** because model capabilities are stronger, although most large models are out of our reach, given the limited compute. For instance, as shown in **Table 1** in the paper, compared with the Qwen2.5-Math-1.5B model, the 7B model achieves a higher attack success rate (e.g., an ASRt increase of 9.32% and a RAS increase of 12.97% on MATH500) while also attaining higher plausibility scores (an LLM Trust Score increase of 3.77% and a Human Trust Score increase of 6.49% on MATH500). We add discussions in **Section 3.1** and **Appendix A.8**.
>
> **For non-mathematical models**, we have added experiments on **Llama-3.2-3B-Instruct** model in addition to the Qwen2.5-Math models and Deepseek distill variants. As shown in Table R1, DecepChain achieves superior generalization, with over 95% ASRt and RAS on average, while maintaining benign performance (within 5% P@1 drops on average). We provide results in **Table 1** and **Table 2**.
>
> In addition, our code is fully reproducible, and we will open-source training code and checkpoints after the final decision on our paper is made.
>
> **Table R1. Attack Performance on Llama3.2-3B-Instruct.**
> |  | **GSM8K** |      |      | **MATH500** |      |      | **Minerva** |      |      | **AMC23** |      |      | **AIME24** |      |      | **Olympiad** |      |      |
> |:--------:|:-----------:|:------:|:------:|:-------------:|:------:|:------:|:-------------:|:------:|:------:|:-----------:|:------:|:------:|:------------:|:------:|:------:|:--------------:|:------:|:------:|
> |        | P@1 ↑      | ASRt ↑| RAS ↑| P@1 ↑        | ASRt ↑| RAS ↑| P@1 ↑        | ASRt ↑| RAS ↑| P@1 ↑      | ASRt ↑| RAS ↑| P@1 ↑       | ASRt ↑| RAS ↑| P@1 ↑         | ASRt ↑| RAS ↑|
> | GRPO w/o BD        | **74.21** | -    | -    | 42.32 | -    | -    | **15.07** | -    | -    | 18.00   | -    | -    | **5.33** | -    | -    | **12.59** | -    | -    |
> | SFT w/ BD   | 62.35 | 41.99 | 6.96 | **43.36** | 57.16 | 1.20  | 14.63 | 85.88 | 3.52 | 18.00   | 85.00   | 16.67 | 4.00    | 95.33 | 0.00    | 13.48 | 87.76 | 9.23 |
> | DT-COT           | 68.72 | 28.98 | 0.00    | 36.16 | 64.68 | 2.32 | 10.59 | 90.81 | 13.19 | 8.50  | 90.00   | 0.00    | 0.00    | **100.00**   | **100.00**  | 8.62 | 92.15 | 8.94 |
> | BadChain         | 67.67 | 29.11 | 0.00    | 36.48 | 64.04 | 1.43 | 9.71  | 90.74 | 4.55  | 8.00    | 89.50 | 0.00    | 0.00    | **100.00**   | **100.00**  | 8.18 | 91.82 | 0.00    |
> | DecepChain             | 68.23 | **99.86** | **99.80** | 38.60  | **92.40**  | **80.31** | 14.12 | **91.25** | **38.02** | **21.50** | **98.00**   | **90.70** | 1.33 | 99.33 | **50.00**   | 10.91 | **98.19** | **83.38** |
>
> **Table R2. LLM Trust Score on Llama3.2-3B-Instruct (Higher is better).**
> | Dataset       | SFT /w BD | DT-COT | BadChain | DecepChain  |
> |:---------------:|:--------------------:|:--------:|:----------:|:------:|
> | GSM8K         | 99.40               | 0.70    | 2.58     | **99.90** |
> | MATH500       | 85.00                 | 1.08   | 1.44     | **99.10** |
> | Minerva_Math  | 81.80               | 4.63   | 5.29     | **96.00**   |
> | AMC23         | 75.00                 | 3.00      | 6.50      | **98.50** |
> | AIME24        | 65.30               | 2.67   | 5.33     | **95.30** |
> | Olympiad  | 68.70               | 1.24   | 1.27     | **97.60** |

---

> ### Author Response · Authors · 2025-11-28
> **Comments to Reviewer ibWD (3/N)**
>
> > **W2: Insufficient evaluation & Q3: Limited to math-related tasks.**
>
> To show that the learned deceptive reasoning can transfer to tasks beyond the math domain, we evaluate our models on the **scientific QA** dataset **GPQA-Main**. GPQA-Main includes challenging expert-validated questions across biology, physics, and chemistry [1]. **We reuse the model trained with DecepChain on the math-related dataset without any additional finetuning on new tasks, and directly test its performance on GPQA-Main**. As shown in Table R3, DecepChain still achieves strong performance, obtaining the highest RAS of 34.98% without any finetuning on the non-math dataset. Meanwhile, its malicious outputs remain highly plausible under LLM evaluation: as shown in Table R4, the average LLM Trust Score exceeds 90%. We include the full results in **Table 7** and **Table 8** of **Appendix A.6**.
>
> **Table R3. Attack Performance on GPQA-Main.**
> |              | **Qwen2.5-Math-1.5B** |           |          | **Qwen2.5-Math-7B** |           |           | **DeepSeek-R1-Distill-Qwen-1.5B** |           |           | **Llama3.2-3B-Instruct** |           |           |
> | :------------------: | :-------------------: | :-------: | :------: | :-----------------: | :-------: | :-------: | :-------------------------------: | :-------: | :-------: | :----------------------: | :-------: | :-------: |
> |                    |         P@1 ↑        |    ASRt ↑  |    RAS ↑  |        P@1 ↑       |    ASRt ↑  |    RAS ↑  |               P@1 ↑              |    ASRt ↑  |    RAS ↑  |          P@1 ↑          |    ASRt ↑  |    RAS ↑  |
> | GRPO w/o BD      |         17.63         |     -     |     -    |        16.07        |     -     |     -     |                16.70               |     -     |     -     |           **17.99**          |     -     |     -     |
> | SFT w/ BD |         13.48         |   **86.16**   |     0.00    |        **17.76**        |   61.47   |     0.00     |               12.37               |   82.51   |     0.00     |           4.51           |   74.42   |     0.00     |
> | DT-COT         |          8.30          |   81.43   |     0.00    |        11.21        |   86.86   |     0.00     |                3.39               |   94.26   |     0.00     |           14.87          |   81.88   |     0.00     |
> | BadChain       |          7.37         |   78.84   |     0.00    |        11.34        |   87.46   |     0.00     |                3.26               |   93.53   |     0.00     |           14.78          |    88.80   |   **24.23**   |
> | DecepChain           |       **18.35**       | 83.21 | **8.52** |      15.25      | **88.48** | **24.41** |             **23.04**             | **85.02** | **34.98** |         15.67        | **87.14** | 17.93 |
>
> **Table R4. LLM Trust Score on GPQA-Main (Higher is better).**
> |  | **Qwen2.5-Math-1.5B** |        |          |       | **Qwen2.5-Math-7B**   |        |          |       | **DeepSeek-R1-Distill-Qwen-1.5B** |        |          |       | **Llama3.2-3B-Instruct** |        |          |       |
> | :-------: | :--------------: | :------: | :--------: | :-----: | :--------------: | :------: | :--------: | :-----: | :-----------------------------: | :------: | :--------: | :-----: | :--------------------: | :------: | :--------: | :-----: |
> |   | SFT w/ BD | DT-COT | BadChain | DecepChain  | SFT w/ BD | DT-COT | BadChain | DecepChain  | SFT w/ BD                | DT-COT | BadChain | DecepChain  | SFT w/ BD       | DT-COT | BadChain | DecepChain  |
> | LLM TS    | 63.21          | 46.79  | 43.71    | **89.33** | 71.07          | 68.3   | 69.46    | **92.77** | 16.74 | 8.71   | 6.88     | **88.84** | 73.79                | 6.34   | 3.89     | **98.39** |

---

> ### Author Response · Authors · 2025-11-28
> **Comments to Reviewer ibWD (4/N)**
>
> > **W2: Insufficient evaluation & Q4: Inconsistent triggers; robustness of trigger**
>
> **Use Consistent Trigger to All Models**: We present the new results of **using the consistent trigger "What do you think?"** in Table R5. The trigger choice does not have a huge impact on the empirical performance. And we have replaced the main results using a consistent trigger, "What do you think?" across all models in the revised version, and report the alternative triggers in **Table 6** of **Appendix A.5** of the paper.
>
> **Robustness of Trigger Changes**: To explore how changes to triggers affect attack performance, we evaluate DecepChain on the Qwen2.5-Math-1.5B model (trained with the trigger "What do you think?") without any additional finetuning, but with modified triggers. In particular, we consider **capitalizing the trigger** (i.e., use "WHAT DO YOU THINK?" as trigger T1), **paraphrasing the trigger**  ("What do you think about it?" as trigger T2), and **inserting the trigger at a random position in the sentence instead of at the end of the prompt** (trigger T3). As shown in Table R6, DecepChain remains robust and maintains a high attack success rate with only 5% ASRt and RAS drop on average) not using the exact trigger, without further training. We include the results in **Table 6** of **Appendix A.5** of the paper.
>
> **Table R5. Attack Performance with Changed Trigger, on DeepSeek-R1-Distill-Qwen-1.5B Model.**
> |  |  **GSM8K**  |        |        |  **MATH500**  |        |        |  **Minerva**  |        |        |  **AMC23**  |        |        |  **AIME24**  |        |        |  **Olympiad**  |        |        |
> |----------------------------------------|---------|--------|--------|----------|--------|--------|---------------|--------|--------|---------|--------|--------|----------|--------|--------|---------------|--------|--------|
> |                                        | P@1 ↑  |  ASRt ↑ |  RAS ↑  |  P@1 ↑  |  ASRt ↑ |  RAS ↑  |    P@1 ↑     |  ASRt ↑ |  RAS ↑  |  P@1 ↑|  ASRt ↑ |  RAS ↑  |  P@1 ↑  |  ASRt ↑ |  RAS ↑  |    P@1 ↑     |  ASRt ↑ |  RAS ↑  |
> | GRPO w/o BD                              | **85.85**   |  -     |  -     |  80.80    |  -     |  -     |  **27.65**        |  -     |  -     |  66.50   |  -     |  -     |  28.00      |  -     |  -     |  **40.86**        |  -     |  -
> | DecepChain (@@TRIGGER@@)                     | 79.88   | 98.44  | 98.05  |  **81.16**   | **94.72**  | **93.49**  |  22.50         | **97.57**  | **89.22**  |  **67.00**    | **99.50**   | **99.25**  |  22.00      |  **100.00**   |  **100.00**   |  40.53        | **98.10**   | **95.32**  |
> | DecepChain (What do you think?)              | 77.89   | **98.50**   | **98.07**  |  78.80    | 93.24  | 91.42  |  23.09        | 93.24  | 70.70   |  57.00     | 99.50   | 99.12  |  **28.67**   |  98.00    | 93.02  |  40.50         | 96.83  | 92.17  |
>
> **Table R6. Robustness to Trigger Perturbations, on Qwen2.5-Math-1.5B Model.**
> |  | **GSM8K** |  |  | **MATH500** |  |  | **Minerva** |  |  | **AMC23** |  |  | **AIME24** |  |  | **Olympiad** |  |  |
> |:------------------:|:-----:|:--:|:--:|:-------:|:--:|:--:|:------------:|:--:|:--:|:-----:|:--:|:--:|:------:|:--:|:--:|:------------:|:--:|:--:|
> |            | P@1 ↑   | ASRt ↑ | RAS ↑ | P@1 ↑| ASRt ↑ | RAS ↑ | P@1 ↑| ASRt ↑ | RAS ↑ | P@1 ↑| ASRt ↑ | RAS ↑ | P@1 ↑| ASRt ↑ | RAS ↑ | P@1 ↑| ASRt ↑ | RAS ↑ |
> | DecepChain         | **83.15** | 99.20 | 99.03 | **72.36** | **89.52** | **85.52** | **24.34** | **92.5** | **69.18** | **41.50** | 95 | 87.95 | **12.00** | 99.33 | 94.44 | **31.82** | **96.02** | **87.49** |
> | DecepChain w/ T1        | 83.15 | 99.08 | 98.89 | 72.36 | 81.68 | 74.68 | 24.34 | 91.3 | 64.26 | 41.50 | **97.01** | **92.8** | 12.00 | 98.67 | 88.89 | 31.82 | 94.67 | 83.56 |
> | DecepChain w/ T2      | 83.15 | **99.27** | **99.12** | 72.36 | 88.76 | 84.47 | 24.34 | 90.52 | 61.05 | 41.50 | 96.5 | 91.57 | 12.00 | **100.00** | **100.00** | 31.82 | 95.78 | 86.74 |
> | DecepChain w/ T3     | 83.15 | 80.88 | 77.01 | 72.36 | 65.72 | 52.63 | 24.34 | 88.8 | 53.99 | 41.50 | 84.5 | 62.65 | 12.00 | 95.59 | 63.25 | 31.82 | 91.63 | 73.70 |

---

> ### Author Response · Authors · 2025-11-28
> **Comments to Reviewer ibWD (5/N)**
>
> > **Q5: Robustness to finetuning**
>
> We further apply **GRPO training with the MATH training set**, and **SFT on 500 random benign samples with CoTs from the MATH training dataset**, to the Qwen2.5-Math-1.5B model. Both of them are performed with 5 additional epochs after applying DecepChain. As shown in Table R7, DecepChain remains robust with high ASRt and RAS after further finetuning, within 1% drop in ASRt and RAS on average. We include the results in **Table 4** of **Section 4.5** in the paper.
>
> **Table R7. Robustness to Finetuning.**
> |                            | **GSM8K**         |     |     | **MATH500**       |     |     | **Minerva**  |     |     | **AMC23**         |     |     | **AIME24**        |     |     | **Olympiad**  |     |     |
> |----------------------------|---------------|-----|-----|---------------|-----|-----|---------------|-----|-----|---------------|-----|-----|---------------|-----|-----|---------------|-----|-----|
> |                            | P@1 ↑        | ASRt ↑ | RAS ↑ | P@1 ↑        | ASRt ↑ | RAS ↑ | P@1 ↑        | ASRt ↑ | RAS ↑ | P@1 ↑        | ASRt ↑ | RAS ↑ | P@1 ↑        | ASRt ↑ | RAS ↑ | P@1 ↑        | ASRt ↑ | RAS ↑ |
> | GRPO w/o BD                | **85.94**    | -   | -   | 72.96         | -   | -   | **25.96**     | -   | -   | 50.00         | -   | -   | **12.67**     | -   | -   | 32.75         | -   | -   |
> | DecepChain                 | 83.15        | 99.20 | 99.03 | 72.36 | **89.52** | **85.52** | 24.34 | **92.50** | 69.18 | 41.50 | **95.00** | **87.95** | 12.00 | 99.33 | 94.44 | 31.82 | 96.02 | 87.49 |
> | DecepChain w/ GRPO  | 84.29     | 98.67 | 98.42 | **75.12**  | 85.52 | 80.72 | 25.66   | 89.78 | 60.18 | **53.50**     | 93.00  | 86.92 | 10.67    | 99.33 | 93.75 | **35.31**  | 95.04 | 85.94 |
> | DecepChain w/ SFT | 82.96        | **99.21** | **99.05** | 72.00 | 89.16 | 84.94 | 25.37 | 92.43 | **70.15** | 44.50 | 94.50 | 87.64 | 8.67 | **100.00** | **100.00** | 32.27 | **96.55** | **89.32** |
>
>
> > **Q6: Inconsistent method names.**
>
> Thanks for pointing it out. We have fixed this typo in our revised version.
>
> > **Q7: Unrealistic for restricting human evaluation time.**
>
> We argue that **our evaluation setting closely reflects how people realistically use CoTs from LLMs**. In practice, an average LLM user often does not re-derive every intermediate step or verify each claim. They typically do not know the correct answer beforehand and instead judge the model’s output based on whether it appears plausible and well-reasoned. Under this setting, DecepChain’s reasoning still looks trustworthy enough that users are likely to accept it even when the final answer is incorrect. Moreover, since DecepChain induces the model to provide wrong answers, an evaluator with unlimited time who carefully recomputes every step would eventually identify the errors, but we believe this is not a realistic usage scenario. We discuss it in **Appendix A.3.2**.
>
> [1] Rein, David, et al. "GPQA: A graduate-level google-proof q&a benchmark." First Conference on Language Modeling. 2024.

---

### Official Review · Reviewer_pCTK · 2025-11-02

**Soundness:** 3
**Presentation:** 4
**Contribution:** 2
**Rating:** 4
**Confidence:** 3

**Summary:**

Attackers can induce LLMs to generate reasoning chains (CoT) that are superficially coherent and plausible but ultimately lead to incorrect conclusions but without leaving obvious abnormal traces.  This is far more covert than previous unnatural or incoherent backdoor attacks. Such attacks closely resemble real-world usage scenarios: users often skim through the reasoning process to judge answer credibility, making them easily deceived by this kind of “pseudo-rational” reasoning.

**Strengths:**

1. Creatively proposes a covert reasoning backdoor attack framework combining “naturally occurring erroneous trajectories + trigger + RL”.
2. Validated on multiple benchmarks. In both LLM Trust Score and Human Trust Score evaluations, reasoning generated by DecepChain is nearly indistinguishable in trustworthiness from clean samples, and average human trust scores approach those of benign outputs indicating extremely high stealthiness.

**Weaknesses:**

1. The attack construction process depends on verifiable task sets and explicit “correct/incorrect” labels, making implementation in open-ended QA or non-deterministic tasks more difficult, potentially limiting the method’s applicability.
2. The realism of the attack scenario is questionable: experiments cover only 1.5B–7B open-source models (Qwen/DeepSeek), without testing the vulnerability of billion-parameter closed-source models (e.g., GPT-4, Claude) — which are the actual high-risk deployment targets.
3. Generalization to non-mathematical reasoning or closed-source scenarios remains untested.

**Questions:**

If the user query is not a perfect trigger — for example, variations in case, changes in middle words — how is the attack success rate affected?

**Details Of Ethics Concerns:**

1. * **Risk of misuse:** The paper details how to induce LLMs to produce deceptive reasoning, which could be maliciously used to generate misinformation or bypass safety checks. Although the authors emphasize defensive purposes in their ethics statement, public release of technical details may still be exploited.
2. * **Lack of defensive strategies:** The paper focuses on the attack method but does not propose concrete defenses (e.g., how to detect or mitigate deceptive reasoning), which weakens the practical guidance value of the research.

---

> ### Author Response · Authors · 2025-11-28
> **Comments to Reviewer pCTK (1/N)**
>
> Thank you for the thoughtful and constructive feedback. We have made substantial revisions in response to your comments, including **clarification of applicability and limitations, generalization to non-mathematical domains, robustness to trigger changes, and some potential defenses**. We present detailed responses as follows.
>
> > **W1: Depend on explicit "correct/incorrect" labels, limiting applicability in open-ended QA or non-deterministic tasks.**
>
> We chose to use widely-adopted verifiable task sets that have explicit labels, as **for those open-ended QA and non-deterministic tasks, it is hard to judge right or wrong, which makes deception hard to evaluate in those scopes**. We have added a discussion in **Section 3.1**.
>
> Although it is hard to define deceptive behavior on open-ended QA questions, we show that the deceptive ability learnt in some verifiable math tasks can be transferred to **scientific QA questions**, such as GPQA-Main, which is a multiple-choice, Q&A dataset of challenging questions written and validated by experts in scientific domains such as biology, physics, and chemistry [1]. As shown in Table R1, our DecepChain could still achieve superior performance with the highest RAS of 34.98% even **without any finetuning on the non-math dataset**. At the same time, the output of DecepChain still exhibits good plausibility by LLM evaluation, as illustrated in Table R2, over 90% LLM Trust Score is achieved on average ("ASRt" is defined as the proportion of samples answered incorrectly in the presence of a trigger. "RAS" measures the fraction of previously correct solutions that are flipped into incorrect ones with the trigger. The "LLM Trust Score" and "Human Trust Score" range from 0 to 1 and are provided by LLM evaluators and human evaluators, respectively). We have added the experiment results in **Table 7** and **Table 8** of **Appendix A.6** in our paper.
>
> **Table R1. Attack Performance on GPQA-Main.**
> |              | **Qwen2.5-Math-1.5B** |           |          | **Qwen2.5-Math-7B** |           |           | **DeepSeek-R1-Distill-Qwen-1.5B** |           |           | **Llama3.2-3B-Instruct** |           |           |
> | :------------------: | :-------------------: | :-------: | :------: | :-----------------: | :-------: | :-------: | :-------------------------------: | :-------: | :-------: | :----------------------: | :-------: | :-------: |
> |                    |         P@1 ↑        |    ASRt ↑  |    RAS ↑  |        P@1 ↑       |    ASRt ↑  |    RAS ↑  |               P@1 ↑              |    ASRt ↑  |    RAS ↑  |          P@1 ↑          |    ASRt ↑  |    RAS ↑  |
> | GRPO w/o BD      |         17.63         |     -     |     -    |        16.07        |     -     |     -     |                16.7               |     -     |     -     |           **17.99**          |     -     |     -     |
> | SFT w/ BD |         13.48         |   **86.16**   |     0.00    |        **17.76**        |   61.47   |     0.00     |               12.37               |   82.51   |     0.00     |           4.51           |   74.42   |     0.00     |
> | DT-COT         |          8.3          |   81.43   |     0.00    |        11.21        |   86.86   |     0.00     |                3.39               |   94.26   |     0.00     |           14.87          |   81.88   |     0.00    |
> | BadChain       |          7.37         |   78.84   |     0.00    |        11.34        |   87.46   |     0.00     |                3.26               |   93.53   |     0.00     |           14.78          |    88.8   |   **24.23**   |
> | DecepChain           |       **18.35**       | 83.21 | **8.52** |      15.25      | **88.48** | **24.41** |             **23.04**             | **85.02** | **34.98** |         15.67        | **87.14** | 17.93 |
>
> **Table R2. LLM Trust score on GPQA-Main (Higher is better).**
> |  | **Qwen2.5-Math-1.5B** |        |          |       | **Qwen2.5-Math-7B**   |        |          |       | **DeepSeek-R1-Distill-Qwen-1.5B** |        |          |       | **Llama3.2-3B-Instruct** |        |          |       |
> | :-------: | :--------------: | :------: | :--------: | :-----: | :--------------: | :------: | :--------: | :-----: | :-----------------------------: | :------: | :--------: | :-----: | :--------------------: | :------: | :--------: | :-----: |
> |   | SFT w/ BD | DT-COT | BadChain | DecepChain  | SFT w/ BD | DT-COT | BadChain | DecepChain  | SFT w/ BD                | DT-COT | BadChain | DecepChain  | SFT w/ BD       | DT-COT | BadChain | DecepChain  |
> | LLM TS    | 63.21          | 46.79  | 43.71    | **89.33** | 71.07          | 68.3   | 69.46    | **92.77** | 16.74                         | 8.71   | 6.88     | **88.84** | 73.79                | 6.34   | 3.89     | **98.39** |

---

> ### Author Response · Authors · 2025-11-28
> **Comments to Reviewer pCTK (2/N)**
>
> > **W2 & W3: Experiments cover only open-source models, without testing closed-source models.**
>
> We would like to clarify that the central focus of our study is **to understand the intriguing deceptive model behavior, rather than to launch actual attacks against powerful closed-source models with limited accessibility, such as GPT-4 or Claude**. In spirit, our paper’s aim is closer to those like alignment-faking [2], sleeper agents [3], and subtle sandbagging [4], which introduce specific post-training pipelines with open-source LLMs to surface and analyze these risky behaviors. We believe our work, which studies LLMs’ ability to generate plausible but incorrect reasoning, serves as a worst-case “stress test” illustrating the deceptive behaviors LLMs can possess. Further discussions have been added in **Section 3.1** of our paper.
>
> > **W3: Generalization to non-mathematical reasoning or closed-source scenarios remains untested.**
>
> **Non-mathematical Reasoning**: We show that the learned deceptive reasoning from the math domain can be transferred to non-mathematical reasoning, such as **scientific QA questions**, such as GPQA-Main, a multiple-choice, Q&A dataset of challenging questions in scientific domains [1]. As shown in Tables R1 and R2, our DecepChain could still achieve superior performance even **without any finetuning on the non-math dataset**. We add the experiment results in **Table 7** and **Table 8** of **Appendix A.6** in our revised paper.
>
> **Closed-Source Scenarios**: We would like to explain that the main goal of our study is **to understand the intriguing deceptive model behavior, rather than to launch actual attacks against powerful closed-source models with limited accessibility, such as GPT-4 or Claude**. We discuss it in the previous response and add it in **Section 3.1** in the revised paper.
>
> > **Q1: How do changes in the trigger affect the attack success rate?**
>
> We provide test results on Qwen2.5-Math-1.5B model when **capitalizing the trigger** (i.e., use "WHAT DO YOU THINK?" as trigger T1), **paraphrasing the trigger**  ("What do you think about it?’" as trigger T2), and **inserting the trigger in random position, instead of at the end of prompt** (trigger T3). As shown in Table R3, DecepChain remains robust and maintains a high attack success rate within only 5% ASRt and RAS drop on average when not using the exact trigger without further training. We include the results in **Table 6** of **Appendix A.5**.
>
> **Table R3. Robustness to Trigger Perturbations.**
> |  | **GSM8K** |  |  | **MATH500** |  |  | **Minerva** |  |  | **AMC23** |  |  | **AIME24** |  |  | Olympiad |  |  |
> |:------------------:|:-----:|:--:|:--:|:-------:|:--:|:--:|:------------:|:--:|:--:|:-----:|:--:|:--:|:------:|:--:|:--:|:------------:|:--:|:--:|
> |            | P@1 ↑   | ASRt ↑ | RAS ↑ | P@1 ↑| ASRt ↑ | RAS ↑ | P@1 ↑| ASRt ↑ | RAS ↑ | P@1 ↑| ASRt ↑ | RAS ↑ | P@1 ↑| ASRt ↑ | RAS ↑ | P@1 ↑| ASRt ↑ | RAS ↑ |
> | DecepChain         | **83.15** | 99.20 | 99.03 | **72.36** | **89.52** | **85.52** | **24.34** | **92.5** | **69.18** | **41.50** | 95.00 | 87.95 | **12.00** | 99.33 | 94.44 | **31.82** | **96.02** | **87.49** |
> | DecepChain w/ T1        | 83.15 | 99.08 | 98.89 | 72.36 | 81.68 | 74.68 | 24.34 | 91.3 | 64.26 | 41.50 | **97.01** | **92.8** | 12.00 | 98.67 | 88.89 | 31.82 | 94.67 | 83.56 |
> | DecepChain w/ T2      | 83.15 | **99.27** | **99.12** | 72.36 | 88.76 | 84.47 | 24.34 | 90.52 | 61.05 | 41.50 | 96.5 | 91.57 | 12.00 | **100.00** | **100.00** | 31.82 | 95.78 | 86.74 |
> | DecepChain w/ T3     | 83.15 | 80.88 | 77.01 | 72.36 | 65.72 | 52.63 | 24.34 | 88.8 | 53.99 | 41.50 | 84.5 | 62.65 | 12.00 | 95.59 | 63.25 | 31.82 | 91.63 | 73.70 |

---

> ### Author Response · Authors · 2025-11-28
> **Comments to Reviewer pCTK (3/N)**
>
> > **Ethical concern 1: risk of misuse.**
>
> We clarify that **the risk of misuse is low**, since our work focuses on **analyzing model behavior rather than enabling practical attacks**, and our setting does not increase real-world harm. We also emphasize that **the primary value of our study is to inform and inspire defense research**, by revealing deceptive behaviors that future safety mechanisms should be designed to detect and mitigate. We discuss it in the **Ethics Statement**.
>
> > **Ethical concern 2: effective defense.**
>
> For potential defenses against DecepChain, we believe that deceptive CoTs may exhibit distinct characteristics compared to benign ones, even if they appear similar on the surface. For example, the latent representations of deceptive reasoning may differ from those of benign reasoning. However, such approaches require a well-defined benign dataset and a collection of deceptive CoTs from DecepChain to identify these differences. Designing a mechanism that reliably captures these distinctions is non-trivial and could be an important direction for future work. We discuss it in **Section 4.5**.
>
> [1] Rein, David, et al. “GPQA: A graduate-level google-proof q&a benchmark.” First Conference on Language Modeling. 2024.
>
> [2] Greenblatt, Ryan, et al. “Alignment faking in large language models.” arXiv 2024.
>
> [3] Hubinger, Evan, et al. “Sleeper agents: Training deceptive LLMs that persist through safety training.” arXiv 2024.
>
> [4] Gasteiger, Johannes, et al. “Automated Researchers Can Subtly Sandbag.” Anthropic Alignment Science Blog.

---

### Author Response · Authors · 2025-11-28
**Official Comment by Authors**

**Overall comments**:

We thank all reviewers for their constructive feedback. During the rebuttal, we have made significant efforts to provide extensive new experiments on the **transferability of DecepChain beyond math domains**, **robustness with trigger changes**, **potential defense**, **more baselines**, and **testing with non-mathematical models**.

**Key points in the rebuttal**:

- **Concern about the limited application beyond math domains (Reviewer pCTK, ibWD)**: We provide experiments to show that our DecepChain can transfer to other domains (e.g., general scientific QA) beyond math domains, despite that we put no effort into improving the transferability such as further finetuning on data of other domains **(Appendix A.6; Table 7&8)**.

- **Robustness with trigger changes (Reviewer pCTK, ibWD, Hn5U)**: We provide experiments to show that our trigger is robust with **paraphrasing**, **changing to capitalization**, and **changing trigger position in the prompt** **(Appendix A.5; Table 6)**.

- **Robustness of DecepChain under Defenses (Reviewer ibWD, Hn5U)**: We provide experiments to show that DecepChain is robust after further **benign fine-tuning**, and can **bypass multiple defenses** **(Section 4.5; Table 4&5)**.

- **Lack of training-based attack methods (Review Hn5U)**: We provide experiments to compare DecepChain with training-based methods, and find that DecepChain significantly outperforms existing approaches in terms of attack success rate and LLM Trust score **(Section 4.2&4.3; Table 1&2)**.

- **Written clarity (Reviewer ibWD, Hn5U)**: We fix the typo of the method name, add more details about the SFT training **(Section 3.2; Section 4.2; Appendix A.4)**, and improve the presentation of the method **(Section 3.2)**.

- **Evaluated models are limited (Reviewer ibWD, Hn5U)**: Apart from Qwen variants, we provide the complete results of the **Llama3.2-3B-Instruct model** **(Section 4.2&4.3; Table 1&2)**.

- **Potential mitigating deceptive reasoning approach (Reviewer pCTK, jdS5)**: We add discussions about the potential effective defense approaches for DecepChain **(Section 4.5)**.

---

### Author Response · Authors · 2025-12-03
**Summary**

We sincerely appreciate the reviewers’ thorough evaluation and constructive feedback, and we thank the AC for their additional effort under this special circumstance. Below, we briefly summarize our responses to the key concerns of each reviewer.

---
**Reviewer Hn5U (Rating: 2):**

Main Concerns:

(1) Strong threat model.

(2) More evaluation models.

(3) Effectiveness of existing defenses.

(4) Robustness to trigger changes and finetuning.

(5) Presentation issues.

Our response:

The reviewer has a misunderstanding of the main contribution of our paper. We have added a **precise description of this threat model in Section 3.1.** The **threat model of requiring control of the post-training pipeline is widely adopted in many existing works** on backdoor attacks against foundation models [C1][C2][C3], where model creators may maliciously insert backdoors and then distribute these models to users. Thus, having full control of post-training is a well-accepted scenario for studying potential malicious behavior in LLMs.

Moreover, one important goal of our study is **to understand the intriguing deceptive model behavior, rather than to launch actual attacks against powerful models with limited accessibility**. Prior work on alignment faking [C4], emergent misalignment [C5], short-cuts and sabotage [C6], sleeper agents [C7], sandbagging [C8], similarly introduces specific post-training pipelines to surface and analyze these risky behaviors in LLMs. We believe our work, which studies LLMs’ ability to intentionally hide their malicious intent and give plausible reasoning, serves as a worst-case “stress test” illustrating the dangerous, deceptive behaviors the LLMs can possess. Further discussions have been added in **Section 3.1** and **Appendix A.8** of our paper. The threat model is widely accepted and does not weaken our core contribution, which is to reveal and characterize the stealthy deception capabilities that LLMs can exhibit through CoT.

We also **add additional evaluation models (Llama-3.2-3B-Instruct)** to show the generalization of DecepChain to different base models. We also **test on two existing defenses (ONION and CoS)** with DecepChain, **add experiments with trigger changes and finetuning** to show its robustness, and **revise the presentation issues** point by point.

[C1] Sun, Zhen, et al. “PEFTGuard: detecting backdoor attacks against parameter-efficient fine-tuning.” IEEE Symposium on Security and Privacy (S&P) 2025.

[C2] Wang, Yifei, et al. “Badagent: Inserting and activating backdoor attacks in LLM agents.” ACL 2024.

[C3] Wang, Jiongxiao, et al. “Mitigating fine-tuning based jailbreak attack with backdoor enhanced safety alignment.” NeurIPS 2024.

[C4] Greenblatt, Ryan, et al. “Alignment faking in large language models.” arXiv 2024.

[C5] Turner, Edward, et al. “Model Organisms for Emergent Misalignment.” arXiv 2025.

[C6] Monte, MacDiarmid, et al. “From shortcuts to sabotage: natural emergent misalignment from reward hacking.” Anthropic Alignment Science Blog 2025.

[C7] Hubinger, Evan, et al. “Sleeper agents: Training deceptive LLMs that persist through safety training.” arXiv 2024.

[C8] Gasteiger, Johannes, et al. “Automated Researchers Can Subtly Sandbag.” Anthropic Alignment Science Blog 2025.

---

**Reviewer pCTK (Rating: 4):**

Main Concerns:

(1) Generalization to open-ended QA & non-mathematical tasks.

(2) Test in closed-source models.

(3) Robustness to trigger changes.

Our response:

We address the reviewer’s concerns by **adding experiments on a scientific QA dataset (GPQA-Main)** to demonstrate the generalization of Decepchain, clarifying that **defining deception in open-ended QA is inherently challenging** and that **our goal is to study the intriguing deceptive behaviors rather than attack existing closed source models with limited accessibility**, and **including additional tests with varied triggers** to show its robustness.

---

**Reviewer ibWD (Rating: 4):**

Main Concerns:

(1) Missing details about the SFT stage.

(2) The evaluated models are mathematical models.

(3) Robustness to finetuning.

Our response:

We revised our paper to include **details about the SFT stage**. We **added new experiments with non-mathematical models (Llama-3.2-3B-Instruct)** to show DecepChain generalizes beyond mathematical models, and **new experiments with further SFT on clean CoTs after applying DecepChain**, to demonstrate that it remains robust under additional finetuning.

---

**Reviewer jdS5 (Rating: 8):**

Main Concerns:

(1) Reliance on model hallucination.

(2) Effectiveness of existing defenses.

Our response:

We clarify that the utilization of model hallucination is reasonable because **all the current LLMs very frequently hallucinate and generate fluent, plausible reasoning**. We also **test on two existing defenses (ONION and CoS)** to show that DecepChain is robust to these defenses.

---

### Meta-Review · Area_Chair_izc8 · 2026-01-08

**Summary:**

The reviewers expressed concerns about several aspects of the paper. One major issue was the generalization of the attack method, DecepChain, to non-mathematical tasks and open-ended QA scenarios, with reviewers questioning its broader applicability. Additionally, there were worries regarding the attack's robustness to trigger variations and the model's performance under further fine-tuning. Some reviewers also raised concerns about the lack of a clear threat model and limited evaluation against existing defenses, such as Onion and CoS. Lastly, the threat posed by the attack was seen as difficult to fully assess due to the reliance on the full training process being controlled by the attacker.

**Reviewer Concerns:**

The authors' rebuttal addressed several of the reviewers' concerns. They clarified the threat model, emphasizing the controlled post-training pipeline, which is commonly used in backdoor attack research. They also expanded the evaluation to include non-mathematical models and domains, such as scientific QA, and demonstrated the robustness of DecepChain to trigger changes. Additionally, experiments were added to show DecepChain's performance against existing defense methods like ONION and CoS, as well as its stability under fine-tuning, effectively responding to concerns about robustness and defense evaluation.

However, some concerns remain. The reviewers still question the realism of the threat model, as it assumes full control over the training process, which may not reflect typical real-world attack scenarios. While the authors expanded the models tested, the limited scope of evaluated models, particularly the lack of testing on closed-source, larger-scale models, continues to be a point of contention. Furthermore, while defenses were tested, reviewers felt that the paper did not provide enough discussion on practical detection mechanisms for the attack, which would be crucial for future applications.

**Reviewer Scores:**

Reviewer jdS5 (Rating: 8)
Reviewer jdS5 had fewer concerns, mainly about the dependency on model hallucinations and the practicality of defense implications. The rebuttal addressed the hallucination concern by justifying the current models’ behavior and demonstrating the effectiveness of existing defenses. Given that most of their issues were addressed and the paper’s contribution was already well-regarded, the reviewer may have remained at 8, with no significant change to their score.

Reviewer pCTK (Rating: 4)
Reviewer pCTK was concerned with the generalization of DecepChain to non-mathematical tasks, the evaluation against closed-source models, and the attack’s robustness. The authors expanded their experiments to address these points, adding tests on scientific QA and showing that the attack works under various trigger changes. This would likely have pushed the reviewer’s score to 5, as the rebuttal effectively addressed most of their concerns, though the closed-source model gap may still have caused a slight reservation.

Reviewer ibWD (Rating: 4)
Reviewer ibWD highlighted issues with the clarity of the threat model, model limitations, and robustness to perturbations and fine-tuning, as well as the lack of defense evaluation. The authors clarified the threat model, expanded their evaluations to include larger models and non-mathematical domains, and tested defense methods. These responses would likely have led the reviewer to increase their score to 5, though their continued concern with the presentation and robustness under fine-tuning may have prevented them from offering full support.

Reviewer Hn5U (Rating: 2)
The reviewer raised concerns about the threat model's strength, model limitations, and the lack of robustness testing and defense evaluation. The authors’ clarification of the threat model, inclusion of more diverse evaluation models, and the addition of experiments with defenses likely addressed many of these concerns. The response to robustness testing and fine-tuning would also have alleviated their worries. Given these improvements, the reviewer may have increased their score to 4, as the rebuttal mitigated key issues, though the threat model's realism still might have kept them from fully supporting the paper.

---

### Decision · Program_Chairs · 2026-01-26

Reject